# Implementing a Multifaceted Intervention among Internal Medicine Residents with Audit and Educative Data Feedback Significantly Reduces Low-Value Care in Hospitalized Patients

**DOI:** 10.3390/jcm11092435

**Published:** 2022-04-26

**Authors:** Omar Kherad, Ezra Bottequin, Dorsaf Steiner, Axelle Alibert, Rodolphe Eurin, Hugo Bothorel

**Affiliations:** 1Internal Medicine Division, Hôpital de la Tour and University of Geneva, 1217 Geneva, Switzerland; 2Business Intelligence Unit, Hôpital de la Tour, 1217 Geneva, Switzerland; ezra.bottequin@latour.ch; 3Quality Department, Hôpital de la Tour, 1217 Geneva, Switzerland; dorsaf.steiner@latour.ch (D.S.); axelle.alibert@latour.ch (A.A.); 4General Management Department, Hôpital de la Tour, 1217 Geneva, Switzerland; rodolphe.eurin@latour.ch; 5Research Department, Hôpital de la Tour, 1217 Geneva, Switzerland; hugo.bothorel@latour.ch

**Keywords:** audit, data feedback, low-value care

## Abstract

Background: The dissemination of recommendations on low-value care alone may not lead to physicians’ behavioral changes. The aim of this study was to evaluate whether a multifaceted behavioral intervention among internal medicine residents could reduce low-value care in hospitalized patients. Methods: A pre–post quality improvement intervention was conducted at the Internal Medicine Division of La Tour hospital (Geneva, Switzerland) from May 2020 to October 2021. The intervention period (3 months) consisted of a multifaceted informational intervention with audits and educative feedback about low-value care. The pre- and post-intervention periods including the same six calendar months were compared in terms of number of blood samples per patient day, prescription rates of benzodiazepines (BZDs) and proton pump inhibitors (PPIs), as well as safety indicators including potentially avoidable readmissions, premature deaths and complications. results: A total of 3400 patients were included in this study; 1095 (32.2%) and 1155 (34.0%) were, respectively, hospitalized during the pre- and post-intervention periods. Patient characteristics were comparable between the two periods. Only the number of blood tests per patient day and the BZD prescription rate at discharge were significantly reduced in the post-intervention phase (pre: 0.54 ± 0.43 vs. post: 0.49 ± 0.60, *p* ≤ 0.001; pre: 4.2% vs. post: 1.7%, *p* = 0.003, respectively). PPI prescription rates remained comparable. Safety indicators analyses revealed no significant differences between the two periods of interest. Conclusions: Our results demonstrate a modest but statistically significant effect of a multifaceted educative intervention in reducing the number of blood tests and the BZD prescription rate at discharge in hospitalized patients. Limiting low-value services is very challenging and additional long-term interventions are necessary for wider implementation.

## 1. Introduction

The Choosing Wisely (CW) campaign has had a substantial reach in mobilizing efforts to reduce low-value care, achieved largely by engaging physician specialty societies in stewardship and producing top-five lists with hundreds of low-value interventions [1,2].

The Swiss Society of General Internal Medicine (SSGIM) launched in 2014 the *smartermedicine* CW campaign to optimize quality and efficiency in the Swiss health system. This campaign published two lists of five low-value interventions to be avoided in Swiss ambulatory internal medicine. In 2016, SSGIM extended the campaign to the hospital setting to create recommendations targeting hospital interventions that have shown to provide little meaningful benefit and present a risk of generating harms and costs. This campaign was led by an expert committee of six hospitalists [3]. The selection process had to take into account the practices and challenges in the Swiss healthcare system and was based on evidence level, frequency, cost savings, risks and benefits for patients. As part of its top five list of low-value interventions that may cause more harm than benefits, the SSGIM has cited the measurement of daily basic blood sample in the absence of clinical changes, the use of benzodiazepines (BZDs) in older adults and the long-term treatment with proton pump inhibitors (PPIs) [4]. These interventions are easy to extract and are frequently considered as low-value in the specialized literature. 

These top-five lists draw attention to low-value services, but the dissemination of these recommendations alone may not lead to physician behavioral changes [5]. These lists should be translated into measurable recommendations and valid quality indicators in the hope of assessing their effect on provider behavior [1,2]. When comparing different approaches to nationwide implementations of CW recommendations, multifaceted interventions with audit and data feedback seem essential to assess whether physicians adhere to the recommendations in their routine clinical practice [6,7,8,9].

Residency training represents a great opportunity to educate physicians about high-value care, to combat rising health care costs and to improve quality of care by eliminating wasteful practices. Purposely, the American College of Physicians (ACP) has launched a high-value care curriculum to encourage cost-effective care implementation in undergraduate and postgraduate education, based on the evidence that early resident education on high-value care may have a sustainable influence [10]. 

The aim of this study was to evaluate whether a multifaceted intervention among internal medicine residents combining performance measurement and comparison feedback could safely reduce these low-value interventions in hospitalized patients.

## 2. Materials and Methods

### 2.1. Setting and Study Design

This prospective pre–post educational study was conducted at La Tour Hospital (Geneva, Switzerland) from May 2020 to October 2021. It represents the 2nd largest hospital in the city, accounting for 2200 admissions per year in its Internal Medicine division. The pre-intervention period (i.e., no intervention) started from 1 May 2020 and ended on 31 October 2020. Because the coronavirus disease 2019 (COVID-19) impacted the patient case mix and our way of prescribing, the period from 1 November 2020 to 31 January 2021 (corresponding to the 2nd wave) was removed from analyses. We then conducted a three-month educational intervention between 1 February and 30 April 2021 and defined the period 1 May 2021 to 31 October 2021 as the post-intervention phase. The pre- and post-intervention periods were defined so that the same calendar months were included to avoid any bias relative to seasonal impacts on patient diagnoses while ensuring a comparable experience of residents. 

The residency program of internal medicine is academically affiliated with the University of Geneva, enrolling 23 residents, 2 junior staffs and 6 senior staffs. The residency program lasts 2 years and physicians were then the same before and after the intervention. All our residents were in their first and second year of training. Our internal medicine division contains 64 acute care hospitals beds distributed among 4 clinical teaching units (CTUs). All residents are integral to patient care delivery and rotate in four-week blocks, supervised by an attending physician (i.e., junior and senior staff), who rotates every 14 days. Laboratory tests and medical prescriptions can be ordered by all resident physicians and staff through the electronic medical record (EMR) (Carefolio, Technology consulting studies, TECOST SA, Fribourg Switzerland). The patient clinical complexity level (PCCL) has been calculated for each treatment episode based on experienced complications and patient comorbidities. The PCCL ranges from 0 (no complication or comorbidity) to 4 (very severe complication or comorbidity) [11].

### 2.2. Intervention

The multifaceted behavioral intervention included: (a) Audit and data feedback. We assessed the prescription behavior of all internal medicine residents working in our CTUs through a dynamic dashboard based on prescription data extracted from our EMR (Figure 1). (b) The residents also received peer comparison feedback. Data were anonymously reported to the prescribers at the individual level permitting peer comparison. The dashboard allowed the resident to compare his/her utilization to the median as well as highest and lowest users among the residents among all 4 CTUs. All residents and staffs had continuous access to the dynamic dashboard, and they were contacted via email from the study team to generate awareness. 

At least once a week, our physician project champion provided an overview of the data; (c) education included weekly quality improvement sessions in small groups focusing on high-value care based on toolkits proposed by the ACP high-value care curriculum. The organization of this weekly one-hour educational session was as follow: first, we created a clinical case scenario based on low-value intervention (i.e., PPIs, BZDs and blood test measurements) that was presented to the group. Participants were invited to give their opinion and behavior in the situation presented without intervention of the moderator. The moderator used variation in care between participants to start the discussion and presented a lecture on the latest recommendations for good clinical practice using a PowerPoint presentation (Appendix A). Data on the variation of low-value services were reported within the group and benchmarking among participants with a reduction target of 20%. The lecture ended with discussion of intervention measures to avoid use of the low-value service within the group (nudge, clinical decision tool). We also used institutional posters that were placed in the medical unit team meeting rooms and directly in the CTUs (Appendix A).

### 2.3. Medical Tests, Procedures and Associated Costs

The medical tests and procedures evaluated in this study were the prescription of BZDs and PPIs as well as blood test measurements. These three interventions were chosen from the Swiss CW top five list given their ease of being reliably extracted as well as their broad applicability. Others intervention from the Swiss top five list were either not accessible or very difficult to analyze without capturing the clinical circumstance.

The proportion of patients with a prescription of BZD and PPI were calculated during the hospitalization and at patient discharge. The proportion of patients with a prescription of BZD and PPI were restricted to patients naive of treatment and aged ≥65 years for BZD as a short course of BZD use may be safe in young adults but in the elderly, even short-term use of benzodiazepines can have dangerous adverse effects.

We have accounted for all prescriptions of new BZDs or PPIs irrespective of the dosage. As we did not capture the clinical context, we were not able to know the reason of prescription neither whether the dosage was increased.

The proportion of patients with a prescription of blood test(s) were calculated during patient hospitalization and the number of prescribed blood tests calculated on a patient day basis to avoid bias on length of stay differences among periods. Since the exact blood test costs could not be retrieved and linked to patient hospitalizations, the global laboratory costs were used as proxy to analyze the intervention effects on costs.

The physician’s prescription rates of BZDs, PPIs and blood tests were also calculated.

### 2.4. Safety Indicators

Safety indicators included potentially avoidable readmissions, premature deaths and global complication scores obtained through the Striving for Quality Level and Analyzing of Patient Expenses (SQLape) software. An avoidable readmission occurs if the patient is readmitted while this was not foreseen at the time of release. A death is defined as premature if it might be prevented with best quality of care in an ideal world. The complications were weighted by scores depending on the context in which they occur: premature death (10), potentially avoidable readmission or reoperation (4), length of stay over expected values (2) and others (1) [12,13,14].

### 2.5. Data Extraction

Raw data extractions from the EMRs were loaded into a dashboard (Power BI, Microsoft Corporation, Redmond, Washington, DC, USA) on a monthly basis. The exports consisted of separate tables for the prescriptions of different types of medical acts (prescription of drugs, laboratory tests). The raw data were processed according to a predefined data model. More specifically, the data from the different raw tables were combined into a single prescription table and the relationship between prescriptions and time was established based on the end date of each medical care episode, i.e., when dashboard users filtered the dashboard by a specific time period, the prescription information would be adjusted to all episodes whose end dates were in the given time frame. The update of the data in the dashboard was performed manually once per month. The dashboard was made available on an internet platform accessible only to hospital staff who had been granted access rights beforehand. It was therefore possible to consult the dashboard at any time of the day with an internet connection and the hospital’s VPN.

### 2.6. Statistical Analyses

Descriptive statistics were used to summarize the data. Continuous variables were reported as the mean ± standard deviation and median (interquartile range (IQR)), and categorical variables were reported as proportions. The normality of continuous variable distributions was assessed by the Shapiro–Wilk test. The significance of differences between periods was determined using the unpaired Student’s *t*-test for normally distributed data and using the Wilcoxon rank sum test for non-normally distributed data. The significance of differences in physician’s prescription rates between the pre- and post-intervention periods was determined using the Wilcoxon signed rank test. For categorical data, the significance of differences in proportion was assessed using the Chi-squared test or the Fisher exact test. In case of prescription differences between the pre- and post-operative periods in the univariate analyses, multivariable logistic/linear regressions were performed to adjust for potential confounding effects related to patient characteristics (i.e., age, sex, PCCL and length of stay). Findings are either presented in odds ratio (OR) or in regression coefficient (beta) with a 95% confidence interval (95% CI). Analyses were performed using R (version 3.6.2, R Foundation for Statistical Computing, Vienna, Austria) and *p*-values < 0.05 were considered significant. 

## 3. Results

A total of 3400 patients were included in this study, among whom 1095 (32.2%) and 1155 (34.0%) were hospitalized respectively during the pre- and post-intervention periods. Regarding patient characteristics (Table 1), the pre- and post-intervention periods were comparable in terms of age (74.1 ± 15.5 vs. 73.3 ± 15.9, *p* = 0.231), sex (52.7% vs. 50.6% of men, *p* = 0.312), PCCL (1.7 ± 1.6 vs. 1.7 ± 1.5, *p* = 0.688) and length of stay (8.9 ± 8.7 vs. 8.8 ± 7.5 days, *p* = 0.291) (Table 1).

### 3.1. Intervention Effects during Hospitalization

During hospitalization, no significant differences could be noted between the two periods in the proportion of patients with a prescription of benzodiazepines (pre: 18.5% vs. post: 15.8%, *p* = 0.143; Figure 2) and PPI (pre: 17.0% vs. post: 16.3%, *p* = 0.652) (Table 1, Appendix A). Likewise, the physician’s prescription rates of benzodiazepines and PPIs did not differ significantly between the two periods (*p* = 0.623 and *p* = 0.384, respectively; Table 1).

The proportion of patients who underwent at least one blood test during the hospital stay was significantly reduced by more than 4% in the post-intervention period (pre: 82.4% vs. post: 78.1%, *p* = 0.011) (Table 1, Figure 3). This significant difference could also be noted on physician’s prescription rate (pre: 82.1% ± 22.2% vs. post: 78.4% ± 21.1%, *p* = 0.035). The multivariate analysis confirmed that prescription of at least one blood test was significantly lower after the intervention compared to the pre-intervention period (OR, 0.68, 95% CI, 0.54–0.85, *p* < 0.001), independently from patient age, sex, length of stay and patient clinical complexity level. Among them, the number of blood tests per hospitalization day was also significantly reduced in the post-intervention phase by 0.05 (pre: 0.54 ± 0.43 vs. post: 0.49 ± 0.60, *p* = <0.001). This result was also confirmed by the multivariate analysis (beta, −0.05, 95% CI, −0.10–0.01, *p* = 0.023). This difference is almost equivalent to a reduction of one blood test every two hospitalizations considering our average length of stay of 9 days (−0.05 × 2 × 9 = −0.9).

### 3.2. Intervention Effects at Patient Discharge

At patient discharge, the proportion of patients with a benzodiazepine prescription was significantly lower after the intervention period (pre: 4.2% vs. post: 1.7%, *p* = 0.003; Figure 2) as opposed to that of PPIs which remained comparable (pre: 9.2% vs. post: 9.4%, *p* = 0.917) (Table 1). The significant reduction in benzodiazepine prescription at patient discharge was also noted on physician’s prescription rate (pre: 4.0% ± 7.5% vs. post: 0.7% ± 2.1%, *p* = 0.025; Table 1).

### 3.3. Safety Endpoints

The safety indicators analyses revealed no significant differences between the two periods of interest in terms of potentially avoidable readmissions (pre: 6.3% vs. post: 5.3%, *p* = 0.434), premature deaths (pre: 2.4% vs. post: 2.1%, *p* = 0.579) or the global complication score (pre: 0.37 ± 1.17 vs. post: 0.30 ± 0.95, *p* = 0.310) (Table 2). 

### 3.4. Intervention Effects on Laboratory Costs

Global laboratory costs decreased from CHF 61.9 ± 70.3 (CHF median, 44.6; IQR, 16.1—84.3) per patient day in the pre-intervention period to CHF 53.1 ± 68.9 (CHF median, 41.5; IQR, 0.0—78.9) in the post-intervention phase (*p* = 0.001), representing an average reduction of CHF 8.8 per patient day. This cost reduction represents CHF 158,400 on a yearly basis for 2000 hospitalizations and an average length of stay of 9 days. 

## 4. Discussion

This study revealed that an intervention utilizing a multifaceted approach of education, data feedback and peer comparison can create safe, significant reductions of low-value care among residents in hospitalized patients. Prescription rates of BZDs at discharge were significantly reduced in the post-intervention phase (pre: 4.2% vs. post: 1.7%, *p* = 0.003) but not for BZD prescription during hospitalization (pre: 18.5% vs. post: 15.8%, *p* = 0.143). The results on BZD prescription seem modest, but even a small reduction in the BZD prescription rate at discharge may have a significant impact in reducing potential serious complications of BZDs such as cognitive impairment, delirium, falls, hip fractures and, possibly, readmissions [15,16,17]. BZD overuse is endemic in Western countries, especially in hospital settings for insomnia disorders. Studies showed that up to 30% of hospitalized patients had at least one BZD prescription in Switzerland [18], and one-third of them (9%) received a repeat BZD prescription at discharge [19]. In our study, 18.5% and 15.8% of patients in the pre- and post-intervention periods, respectively, received at least one BZD prescription during their hospitalization. However, it is worth noting that we only included patients aged >65 years naive of BZD at admission, which may underestimate the prescription rates and preclude conclusions on deprescription, which is a very challenging process with a risk of withdrawal symptoms [20].

The impact of our intervention on blood sample measurement is more relevant, as it resulted in a significant lower number of blood tests per patient hospitalization day (pre: 0.54 ± 0.43 vs. post: 0.49 ± 0.60, *p* ≤ 0.001). Excessive diagnostic blood sample measurement is associated with adverse patient outcomes such as hospital-acquired anemia [21,22], increased blood transfusions, prolonged hospitalization with a subsequent risk of overdiagnosis and possible mortality [23]. Furthermore, reducing the number of blood sample measurements increases the number of days free of blood tests, reducing pain and early-morning awakening for blood draws that are of major disturbance to patients’ quality of life during their hospitalization. 

The mitigation of blood measurements presents an ideal opportunity for improvement of patient care but also for cost-savings. Even though our cost saving of CHF 8.8 per patient day (or US dollars 9.5) was calculated on all laboratory tests, our results are consistent with those of Thakkar et al., who reported a reduction in blood test costs of US dollars 6.3 per patient day following two months of educational intervention [24]. 

The PPI prescription rate did not differ before and after the intervention. A first explanation is certainly the degree of confidence in these recommendations. Indeed, PPI prescriptions often fall into a grey area for which the balance between benefits and harms varies substantially among patients and are backed by little evidence to help decide which patients may benefit [25]. Still, PPIs are among the most widely prescribed drugs in hospitals, and more than half of the indications for prescriptions are unjustified [26]. The lack of data on clinical contexts and prescription indications in our study preclude any conclusions. Continuous efforts are necessary to reduce PPI overuse, especially in the long-term.

Measuring the impact of efforts to eliminate low-value care requires a variety of approaches. Simply informing physicians to order tests parsimoniously in itself is insufficient to drive practice change which requires more robust implementation strategies in regard to the complexities of different practice environments such as the hospital setting [27,28]. A large body of literature evaluated the outcome of interventions aimed at reducing low-value care [8,9]. Most interventions were found to be effective, with multifaceted interventions being more effective and frequently reported compared to single-component interventions [8]. Education programs, patient education, clinical decision support, shared decision making and economic incentives are a few examples. Among these interventions, audit and feedback constitute an approach that is used to improve practice, involving measurement of an individual’s or group’s practice and comparison with standard references or targets. The feedback component may help physicians to adjust their practice when their performance is inconsistent with the group or desired target, as demonstrated by different antimicrobial stewardship studies [29,30]. A recent investigation revealed that a multifaceted intervention using education and data feedback with goal setting and peer comparison can also reduce unnecessary daily blood test measurements on inpatient general medicine, corroborating previous studies [31,32,33]. 

There is increasing interest in use of behavioral science to affect practice in medicine [34]. Audit with educative feedback can be used to nudge physicians’ behaviors [35,36]. Nudge strategies have been suggested as one way to influence habitual behavior, by targeting the subconscious routines and biases that are present in physician behavior [34]. Our pragmatic quality improvement study combined education and feedback methods with competition using the practices of peers to establish a norm. We hypothesized to influence prescribing behavior primarily by increasing physicians’ intentions to appropriately adjust their prescribing after comparison to others performance. Peer comparison might have led residents to make judicious prescribing part of their professional self-image. Our results are in line with previous studies confirming this type of nudge can lower low-value service in hospitalized patients [34]. 

Our study focused on the practice of young residents in internal medicine by using the ACP tool kit and vignettes as a basis for specific education in small groups [10]. Provision of high-value care should be a milestone in physician training and young residents can be good stewards of limited health care resources [10]. Of note, the problem of low-value care is most acute among young physicians, as studies have shown that more experienced physicians practice at lower cost. [37] Increasing knowledge about high-value care among residents and medical students has been associated with reducing inappropriate health care delivery. Furthermore, residents who trained in high-spending regions tend to have higher mean spending compared with those who trained in low-spending regions [38,39]. This underlines the importance of medical education in all teaching hospitals and for faculty as role models for appropriate behaviors. Although the importance of high-value care education is increasingly recognized, there is a lag in implementation with only few residency programs with formal curriculum in high-value care. Our study may serve as an example that can be adapted at other institutions to implement site-specific high-value care initiatives. We hope that following a Plan–Do–Study–Act approach, the audit with data feedback will be able to change the behavior of reluctant physicians in future steps of this quality initiative

### Limitations

The present study has several limitations. First, we did not include a medical unit as a control group in our investigation, and it remains possible that diagnoses at hospital admission differed among the two periods of interest. Furthermore, the use of a control group in our hospital would have been complicated due to the fact of its relatively small size and risks of disseminating the intervention effects beyond the experimental units. We nevertheless demonstrated comparable patient characteristics between the two periods in terms of age, gender, length of stay and patient clinical complexity level, and we performed multivariable analyses where feasible to adjust for potential confounding effects due to the aforementioned patient characteristics. Unfortunately, such analyses could not be performed for benzodiazepine prescription at patient discharge, since the number of concerned patients was insufficient. Second, our data extracted from the EMR do not adequately capture the clinical circumstances that led to ordering a service, which may be essential for some recommendations such as PPI prescription. Therefore, we cannot affirm these interventions were of low-value for all individual patients. Third, we were unable to accurately retrieve blood test costs from the global laboratory expenses. However, we estimated that blood tests accounted for an important proportion of laboratory tests, thereby validating the use of global laboratory costs as a proxy. Fourth, further studies with a greater cohort size would be needed to statistically evaluate differences in terms of safety endpoints. Moreover, the prescription rate might be lower in summer compared to other periods. Since our pre- and post-intervention periods only covered 6 months each (from May to October), future studies would be interesting to further evaluate the impact of such educative intervention on a one-year period. Finally, our intervention was performed at a single hospital medical teaching unit and may not be representative of all hospitals, limiting the generalizability of our findings because local factors likely play an important role. 

## 5. Conclusions

Limiting low-value services is very challenging. Our results demonstrate a modest but statistically significant effect of a multifaceted educative intervention in reducing number of blood tests per patient day and the BZD prescription rate at discharge among internal medicine residents in hospitalized patients. Audit and educative feedback on their prescribing behavior gave them a basis for comparing their practice with the group and for adjusting their practice when their performance was inconsistent. This study provides a starting point for further evaluation of the influence of the initiative on changing behavior by analyzing changes in volume and variation of low-value services. On a larger scale, this study resulted in an increase in the institutional awareness of high-value care principles in general. Future audit and feedback-based interventions focusing on other physician groups and different low-value interventions or quality indicators could be undertaken using this study method.

## Figures and Tables

**Figure 1 jcm-11-02435-f001:**
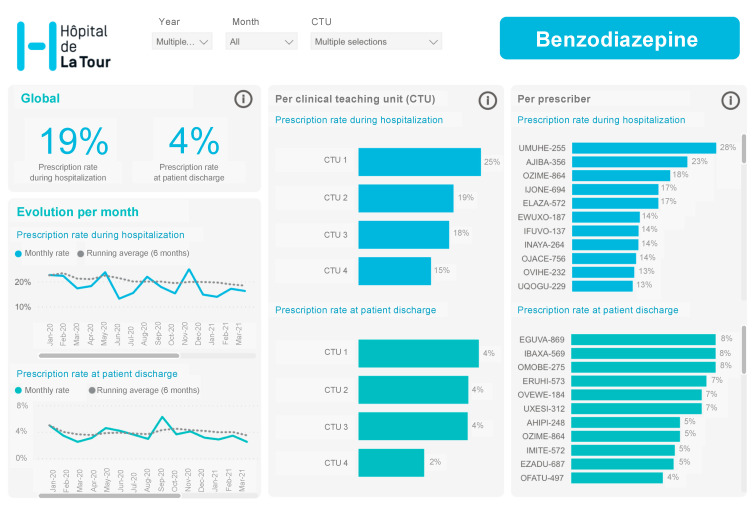
Dynamic dashboard based on prescription data.

**Figure 2 jcm-11-02435-f002:**
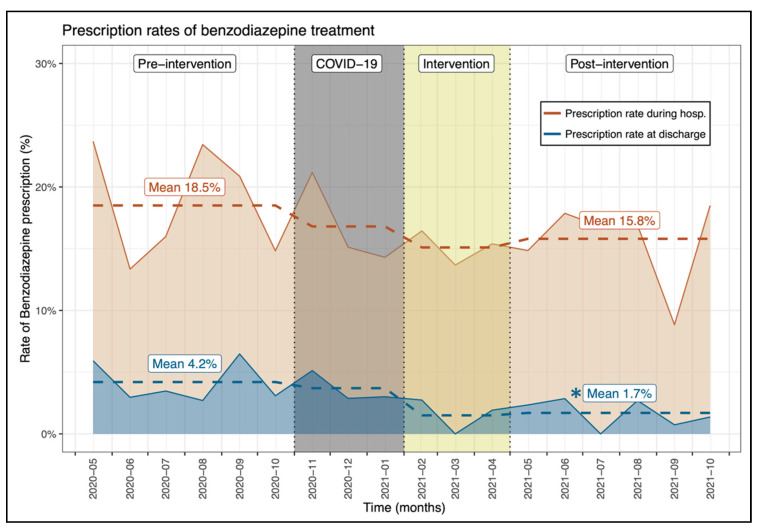
Evolution of the proportion of patients with a benzodiazepine prescription between the investigated periods. * Indicates a significant difference between the pre- and post-intervention prescription rate.

**Figure 3 jcm-11-02435-f003:**
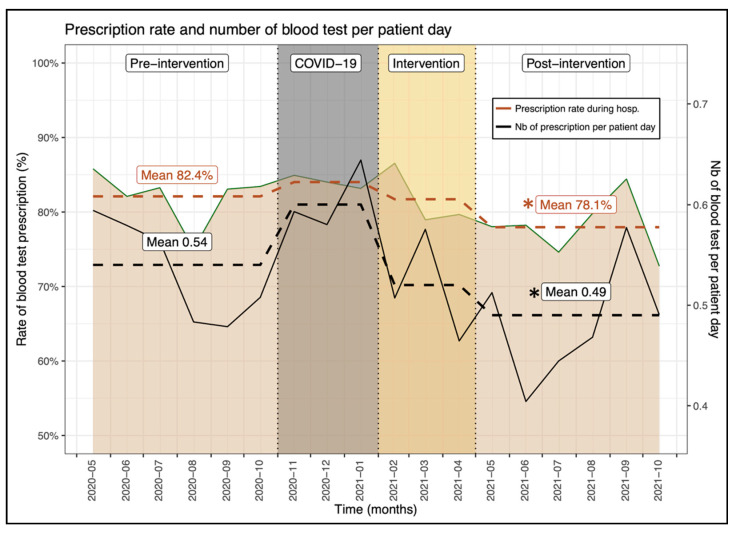
Evolution of the proportion of patients with a blood test(s) prescription between the investigated periods. * Indicates a significant difference between the pre- and post-intervention prescription rate.

**Table 1 jcm-11-02435-t001:** Prescription differences between the pre- and post-intervention periods.

	Pre-Intervention Period (1095 Patients)			Post-Intervention Period (1155 Patients)			*p*-Value
	N (%)			N(%)			
	$\bar{x}$ ± SD	Med	(IQR)	$\bar{x}$ ±SD	Med	(IQR)	
**Patient characteristics**							
Age	74.1 ± 15.5	77.5	(67.0–85.0)	73.3 ± 15.9	77.0	(65.0–84.0)	0.231 *
PCCL	1.7 ± 1.6	2.0	(0.0–3.0)	1.7 ± 1.5	2.0	(0.0–3.0)	0.688 *
Length of stay (days)	8.9 ± 8.7	6.7	(3.3–11.2)	8.8 ± 7.5	7.0	(3.9–11.2)	0.291 *
Age ≥ 65 y.o	843 (77.0)			835 (72.3)			0.185
Male sex	577 (52.7)			584 (50.6)			0.312
**During hospitalisation**							
Patients with prescription							
Benzodiazepine (≥65 y.o)	156 (18.5)			132 (15.8)			0.143
Proton Pump Inhibitor	186 (17.0)			188 (16.3)			0.652
Blood test	902 (83.1)			902 (78.1)			0.011
Phys. prescription rate (%) ⟟							
Benzodiazepine (≥65 y.o)	15.4 ± 17.6	14.6	(0.0–22.6)	11.7 ± 11.8	12.5	(0.0–19.1)	0.623 ⋄
Proton Pump Inhibitor	14.6 ± 14.3	13.5	(13.5–0.0)	16.4 ± 20.6	15.8	(0.0–21.3)	0.384 ⋄
Blood test	82.1 ± 22.2	83.6	(81.2–100.0)	78.4 ± 21.1	80.8	(80.8–72.3)	0.035 ⋄
Blood tests per patient day **	0.54 ± 0.43	0.42	(0.26–0.69)	0.49 ± 0.60	0.37	(0.20–0.57)	<0.001 *
**At patient discharge**							
Patients with prescription							
Benzodiazepine (≥65 y.o)	35 (4.2)			14 (1.7)			0.003
Proton Pump Inhibitor	101 (9.2)			108 (9.4)			0.917
Phys. prescription rate (%) ⟟							
Benzodiazepine (≥65 y.o)	4.0 ± 7.5	0.0	(0.0–5.5)	0.7 ± 2.1	0.0	(0.0–0.0)	0.025 ⋄
Proton Pump Inhibitor	8.6 ± 11.9	5.4	(0.0–9.3)	6.0 ± 6.2	5.7	(0.0–9.3)	0.394 ⋄

IQR, Interquartile range; PCCL, Patient clinical complexity level; y.o, year old; Med, Median; $\bar{x}$, Mean; SD, Standard deviation; Phys. Physician’s; Underlined values indicate significant *p*-values (<0.05); * *p*-value calculated using Wilcoxon rank sum tests; ** Among patients with the concerned prescription; ⋄ *p*-value obtained using Wilcoxon signed rank tests; ⟟ The mean physician’s prescription rates slightly differ from the proportions of patients with prescription since each physician did not treat the same number of patients over the period of interest.

**Table 2 jcm-11-02435-t002:** Comparison of safety endpoints between the pre- and post-intervention periods.

	Pre-Intervention (n = 1095 Patients)			Post-Intervention (n = 1155 Patients)			*p*-Value
	N (%)			N(%)			
	$\bar{x}$ ± SD	Med	(IQR)	$\bar{x}$ ± SD	Med	(IQR)	
**Pot. avoid. readmission**							
Eligible hospitalisations	656			685			
Readmission rate (%)	41 (6.3%)			36 (5.3%)			0.434
Readmission delay (days)	11.9 ± 10.1	8.0	(3.0–21.0)	14.3 ± 8.6	14.0	(6.5–22.3)	0.160 *
**Premature deaths**							
Eligible hospitalisations	992			1069			
Death rate (%)	24 (2.4%)			22 (2.1%)			0.579
**Complications**							
Eligible hospitalisations	1056			1133			
Global complication score	0.37 ± 1.17	0.00	(0.00–0.00)	0.30 ± 0.95	0.00	(0.00–0.00)	0.310 *

All indicators were calculated using the SQLape software; IQR, Interquartile range; Pot. avoid., Potentially avoidable; Med, Median; $\bar{x}$, Mean; * *p*-value calculated using Wilcoxon rank sum tests.

## Data Availability

The data presented in this study are available upon request from the corresponding author.

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
