# Peer review of "Implementing a Multifaceted Intervention among Internal Medicine Residents with Audit and Educative Data Feedback Significantly Reduces Low-Value Care in Hospitalized Patients"

_jcm, 2022, doi:10.3390/jcm11092435_

Round 1

Reviewer 1 Report

Thank you for giving the opporutnity to review this work on the evalaution of low-value care in internal medecine patients. 

The authors show that, among the evaluated low-value care, blood test prescription per day and BZD at discharge were reduced between the two periods in univariate analysis. 

This is interesting. However, I have a couple of comments and questions. 

1) Could you introduire the CW top-five list ? 

2) Why did you choose the prescription rate of BZD and PPI and blood measurements among the top-five ?

3) Was it the same physicians evaluated in the pre and post educational intervention ?  

4) Do you think that the fact physician were informed about their prescription rate could impact on their behaviour independently of the 3-month educational intervention?   

5) Why did you only assess  prescription rates of BZD in patients naïve of treatment and aged ≥ 65 years ? 

6) Median is usually given with IQR and not as median (min-max). 

7) In table 1 : you do not provide with the data of blood test at discharge, is it intentional ? 

8) During the summer, the prescription rate is lower. Could you explain why ? does it results from paramedic holidays ? or reduced activity during the summer period ? 

9) I would suggest to temper the conclusion. You can’t state that your intervention results in significant reduction of low-value care as) no multivariate analysis to adjust for confusion variables is performed and 2) not all prescriptions were reduced after intervention.  

10) could you provide more details concerning the 3 months interventions ?

11) do you have the data on resident experience ? In what year of residency are they ?

Author Response

1) Could you introduce the CW top-five list ? 

The Swiss CW top list has been introduced in the introduction section.

Line 76 “The Swiss Society of General Internal Medicine (SSGIM) launched in 2014 the smartermedicine CW campaign, in order to optimize quality and efficiency in the Swiss health system. This campaign published two lists of five low-value interventions to be avoided in Swiss ambulatory internal medicine. In 2016, SSGIM extended the campaign into the hospital setting, to create recommendations targeting hospital interventions that have shown to provide little meaningful benefit and present a risk of generating harms and costs. This campaign was led by an expert committee of six hospitalists. The selection process had to take into account the practices and challenges in the Swiss healthcare system and was based on evidence level, frequency, cost savings, risks and benefits for patients. As part of its top five list of low-value interventions that may cause more harm than benefits, the SSGIM has cited the measurement of daily basic blood sample in the absence of clinical changes, the use of benzodiazepines (BZD) in older adults and the long-term treatment with proton pump inhibitors (PPI). These interventions are easy to extract and are frequently considered as low-value in the specialized literature. “

2) Why did you choose the prescription rate of BZD and PPI and blood measurements among the top-five ?

Some low-value services reported in the Swiss CW top five lists were not selected because measurement from our EMR was not accessible or because the low-value service was very difficult to analyze such the 4th item: “Don’t let older adults lie in bed during their hospital stay”. Furthermore, our analysis based on EMR data did not adequately capture the clinical circumstances that led to ordering a service, which may be essential for some recommendations such as antibiotic of upper respiratory tract infection.

PPI, BZD and blood measurements are medical interventions that are frequently reported as low-value in the hospital setting, having shown to provide little meaningful benefit and present a risk of generating harms and costs. Furthermore, even if we do not have the clinical context of the prescription of those 3 items, an accepted alternative means of assessing physicians’ use of low-value care is to identify variations in medical practice between physicians. Those data were easy to extract and fitted perfectly for this comparison.

We added a sentence line 186: “Others intervention from the Swiss top five list were either not accessible or very difficult to analyze without capturing the clinical circumstance”.

3) Was it the same physicians evaluated in the pre and post educational intervention ? 

Yes, our residency program lasts 2 years and physicians were then the same before and after the intervention. This has been specified in the methods section line 129.

“The residency program lasts 2 years and physicians were then the same before and after the intervention”

4) Do you think that the fact physician were informed about their prescription rate could impact on their behaviour independently of the 3-month educational intervention?   

Yes, we hope that the data measurement and feedback by itself may change the physician behavior. We hope that following a Plan-Do-Study-Act approach, we will be able to change the behavior of reluctant physicians in future steps of this quality initiative

We added a sentence line 389. We hope that following a Plan-Do-Study-Act approach, the audit with data feedback will be able to change the behavior of reluctant physicians in future steps of this quality initiative

5) Why did you only assess  prescription rates of BZD in patients naïve of treatment and aged ≥ 65 years ? 

The objective of the study was to limit the new prescription of low-value interventions. As mentioned in the text,  our data preclude conclusions on deprescription which is very challenging process. Furthermore, neither BZD nor PPI can be interrupted abruptly without a risk of withdrawal syndrome and rebound effect respectively.

In young adults, a short course of benzodiazepine use may be safe. In the elderly, however, even short-term use of benzodiazepines can have dangerous adverse effects, as reported by the American Geriatrics Society that placed benzodiazepines on a list of medications that should be avoided in patients over 65 years of age.

6) Median is usually given with IQR and not as median (min-max). 

We corrected the manuscript and tables in order to report median and interquartile range. (Lines 291-3 + Tables 1 and 2)

7) In table 1 : you do not provide with the data of blood test at discharge, is it intentional ? 

We are not sure to understand this question. We analyzed data for PPI and BZD at discharge to verify whether patients left the hospital with a prescription. For the blood tests, we have accounted for all the blood analyses during the hospital stay.

8) During the summer, the prescription rate is lower. Could you explain why ? does it results from paramedic holidays ? or reduced activity during the summer period ? 

The summer did not significantly or relevantly differ from other periods in terms of PCCL (1.7±1.5 vs 1.6±1.5, p=0.160) patient age (73.2±15.3 vs 73.9±15.7, p=0.039), length of stay (8.5±8.0 vs 8.8±7.3, p=0.015) or gender (51.4% vs 50.6%, p=0.654). We added in the limitation section: “The prescription rate might be lower in summer compared to other periods. Since our pre and post-intervention periods only cover 6 months each from May to October, future studies analyzing complete year periods would be interesting to further evaluate the impact of such educative intervention. ” Lines 411-5

9) I would suggest to temper the conclusion. You can’t state that your intervention results in significant reduction of low-value care as) no multivariate analysis to adjust for confusion variables is performed and 2) not all prescriptions were reduced after intervention.  

We agree with the reviewer our conclusion is too strongly worded. The conclusion has been tempered and a new sentence has been added line 371 “This study provides a starting point for further evaluation of the influence of the initiative on changing behavior by analyzing changes in volume and variation of low-value services.”

We additionally performed a multivariate logistic regression, which confirmed that the prescription of at least one blood test was significantly lower in the post-intervention period (compared to the pre-intervention period), independently from patient age, sex, length of stay and patient clinical complexity level. Among patients with at least one blood test, a multivariate linear regression confirmed that the number of blood test per patient day was significantly lower in the postoperative period, independently from aforementioned patient characteristics. (Lines 265-71)

Unfortunately, the number of patients concerned by a benzodiazepine prescription at discharge was relatively small which prevented us from performing further analyses. (Lines 402-404)

10) could you provide more details concerning the 3 months interventions ?

The 3-month intervention period is described in method section, line 151. We do agree that we should give more information on the methodology of education and the content of education has been added in the supplementary material.

New sentences have been added Line 162: “The organization of this weekly one-hour educational session was as follow: first, we created a clinical case scenario based on low-value intervention (PPI, BZD and blood tests measurement) that was presented to the group. Participants were invited to give their opinion and behavior in the situation presented, without intervention of the moderator. The moderator used variation in care between participants to start the discussion and presented a lecture on the latest recommendations for good clinical practice using PowerPoint presentation (Supplementary material 1). Data on variation of low-value services were reported within the group and benchmarking among participants with a reduction target of 20%. The lecture ended with discussion of intervention measures to avoid use of the low-value service within the group (nudge, clinical decision tool). We also used institutional posters that were placed in the medical unit team meeting rooms and directly in the CTU. (Supplementary material 2)

11) do you have the data on resident experience ? In what year of residency are they ?

All residents were in their first year (PG1) and second year (PG2) of residency without statistical difference in their prescription behavior. A new sentence has been added in the methods section line 130. “All our residents were in their first and second year of training”.

Reviewer 2 Report

Addressing the issue of overuse in healthcare is an important challenge. The authors demonstrated non-inferiority in patient outcomes when low-value care was reduced by providing dynamic education to residents that included reporting of actual clinical practices over time. This is a clear study of a proactive educational intervention, but the interpretation of the results should be considered. Comments on the manuscript are as follows.

MAJOR COMMENTS

  • The selection of three interventions from the top 10 low-value care:
    The authors selected three interventions based on the ease of being reliably extracted and the broad The number of blood tests should be considered an objective indicator, however, the validity of the interpretation should be carefully evaluated with regard to the prescription of BZD and PPI. Patients who were receiving BZD or PPI prior to admission are not included in Table 1, but an appropriate definition is needed for these patients, such as adding a new sedative or increasing the dosage. Furthermore, is BZD the first choice for insomnia disorders in the local clinical practice? If there is an institutional protocol for the treatment of insomnia that includes the use of other sedatives, it should be presented.
  • Impact of the Intervention
    As the authors implied the possibility of seasonal variation in diagnosis on hospital admission (L71-74), the figures seem to indicate seasonal variation in prescriptions for BZD and PPI. As for the impact of interventions on outcomes, it may be necessary to adjust for the potential confounders such as patient characteristics in multivariate analysis in addition to univariate analysis.

MINOR COMMENTS

  • The results section of the Abstract does not include the results for PPI, which is listed as a considered item in the Methods. There were no significant differences in the use of PPI in the pre- and post-periods, and the appropriate conclusion should include that fact.
  • Materials and Methods:
    Although the methodology of education is described, the content of education should also be presented. Details such as frequency targets for low-value intervention defined by the authors should be presented.
  • L75-82:
    Were all prescriptions and blood tests ordered at the discretion of the residents themselves? Or, does this method mean that all doctors in the Division of Internal Medicine were similarly educated?
  • I couldn’t find the file entitled “supplementary material 1” in the submission. There are 3 files starting with “HDTL_Affiche_” written in French, are they “supplementary material 1”??
  • L115-7:
    Definitions of safety indicators (potentially avoidable readmissions, premature deaths and global complication scores) should be provided.
  • L133-138
    Different analyses are used for normally and non-normally distributed variables. The tables list both mean and median values, and are unable to determine whether the results were obtained using the Student t test or the Wilcoxon rank sum test.
  • L145-148
    Patient characteristics such as age, gender, and PCCL should be described in Table 1. In addition, the diagnosis at hospital admission would be necessary information. In particular, patients with COVID-19 may receive different care compared to the rest of the general patients, as access to the patient may be more difficult.
  • L157:
    The proportion of patients who underwent at least one blood test “during hospital stay”?
  • L161,2 and L181,2
    How were the results calculated?
  • Table1: The P-value for the "≥65 years old" is not presented.
  • L268-270
    The present study considered the reduction of low-value care. The authors discussed site-specific high-value care, however, is this a discussion based on the present study?

Author Response

MAJOR COMMENTS

  1. The selection of three interventions from the top 10 low-value care:
    The authors selected three interventions based on the ease of being reliably extracted and the broad The number of blood tests should be considered an objective indicator, however, the validity of the interpretation should be carefully evaluated with regard to the prescription of BZD and PPI. Patients who were receiving BZD or PPI prior to admission are not included in Table 1, but an appropriate definition is needed for these patients, such as adding a new sedative or increasing the dosage. Furthermore, is BZD the first choice for insomnia disorders in the local clinical practice? If there is an institutional protocol for the treatment of insomnia that includes the use of other sedatives, it should be presented
    .

We thank the reviewer for this very interesting comment. Some low-value services reported in the Swiss CW top five lists were not selected because measurement from our EMR was not accessible or because the low-value service was very difficult to analyze such the 4th item: “Don’t let older adults lie in bed during their hospital stay”.  Furthermore, our analysis based on EMR data did not adequately capture the clinical circumstances that led to ordering a service, which may be essential for some recommendations such as antibiotic of upper respiratory tract infection. We added a sentence line 165: “Others intervention from the Swiss top five list were either not accessible or very difficult to analyze without capturing the clinical circumstance”.

We do agree that clinical circumstances are also critic for the validity of the interpretation for PPI and BZD. PPI, BZD are medical interventions that are frequently reported as low-value in the hospital setting, having shown to provide little meaningful benefit and present a risk of generating harms and costs. Even if we do not have the clinical context of the prescription of those items, an accepted alternative means of assessing physicians’ use of low-value care is to identify variations in medical practice between physicians. Those data were easy to extract and fitted perfectly for this comparison in this study.

We excluded data with prior PPI or BZD prescription as mentioned in the text. The objective of the study was to limit the new prescription of low-value interventions. Our data preclude conclusions on deprescription which is very challenging process. Furthermore, neither BZD nor PPI can be interrupted abruptly without a risk of withdrawal syndrome and rebound effect respectively.

We added a sentence in the methods section line 192 regarding the definition of PPI or BZD prescription: “We have accounted for all prescriptions of new BZD or PPI irrespective of the dosage. As we did not capture the clinical context, we were not able to know the reason of prescription neither whether the dosage was increased”.

We do not have an institutional protocol for the treatment of insomnia. During the 3-months intervention, we gave lecture on the risk of BZD for insomnia in the elderly (supplementary material 1). As an alternative of insomnia, we suggested melatonin, but the decision was left to the discretion of the attending physician.

  • Impact of the Intervention
    As the authors implied the possibility of seasonal variation in diagnosis on hospital admission (L71-74), the figures seem to indicate seasonal variation in prescriptions for BZD and PPI. As for the impact of interventions on outcomes, it may be necessary to adjust for the potential confounders such as patient characteristics in multivariate analysis in addition to univariate analysis.

We performed different multivariate analyses to adjust our findings on blood tests according to patient age, sex, patient clinical complexity level and length of stay. (Lines 265-71)

Since the periods of interest include the exact same calendar months, we did not include the seasons in the multivariate analyses to not disturb the models. We additionally added in the limitations “The prescription rate might be lower in summer compared to other periods. Since our pre and post-intervention periods only cover 6 months each (from May to October), future studies would be interesting to further evaluate the impact of such educative intervention on a one-year period. (lines 411-415)”

MINOR COMMENTS

  1. The results section of the Abstract does not include the results for PPI, which is listed as a considered item in the Methods. There were no significant differences in the use of PPI in the pre- and post-periods, and the appropriate conclusion should include that fact.

We agree with the reviewer and the abstract and conclusion were modified accordingly.

  1. Materials and Methods:
    Although the methodology of education is described, the content of education should also be presented. Details such as frequency targets for low-value intervention defined by the authors should be presented.

The 3-month intervention period is described in method section, line 151. We do agree that we should give more information on the methodology of education and the content of education has been added in the supplementary material 1.

New sentences have been added Line 162: “The organization of this weekly one-hour educational session was as follow: first, we created a clinical case scenario based on low-value intervention (PPI, BZD and blood tests measurement) that was presented to the group. Participants were invited to give their opinion and behavior in the situation presented, without intervention of the moderator. The moderator used variation in care between participants to start the discussion and presented a lecture on the latest recommendations for good clinical practice using PowerPoint presentation (Supplementary material 1). Data on variation of low-value services were reported within the group and benchmarking among participants with a reduction target of 20%. The lecture ended with discussion of intervention measures to avoid use of the low-value service within the group (nudge, clinical decision tool). We also used institutional posters that were placed in the medical unit team meeting rooms and directly in the CTU. (Supplementary material 2)

  1. L75-82:
    Were all prescriptions and blood tests ordered at the discretion of the residents themselves? Or, does this method mean that all doctors in the Division of Internal Medicine were similarly educated?

Yes the blood test are ordered at the discretion of the residents themselves. All residents were in their first year (PG1) and second year (PG2) of residency without statistical difference among their prescription behavior. A new sentence has been added in the methods section line 130. “All our residents were in their first and second year of training”.

  1. I couldn’t find the file entitled “supplementary material 1” in the submission. There are 3 files starting with “HDTL_Affiche_” written in French, are they “supplementary material 1”??

Yes those are supplementary material 1. We also added the PowerPoint presentations in the supplementary file.

  1. L115-7:
    Definitions of safety indicators (potentially avoidable readmissions, premature deaths and global complication scores) should be provided.

Modified as requested and references added. (Lines 203-8)

  1. L133-138
    Different analyses are used for normally and non-normally distributed variables. The tables list both mean and median values, and are unable to determine whether the results were obtained using the Student t test or the Wilcoxon rank sum test.

Specified in tables.

  1. L145-148
    Patient characteristics such as age, gender, and PCCL should be described in Table 1. In addition, the diagnosis at hospital admission would be necessary information. In particular, patients with COVID-19 may receive different care compared to the rest of the general patients, as access to the patient may be more difficult.

Patient characteristics added in Table 1. Since a considerable number of diagnoses at hospital admission exist, we were unable to scientifically evaluate the difference in diagnoses between the two periods (Added in the limitations lines 395-6) with our sample size.

  1. L157:
    The proportion of patients who underwent at least one blood test “during hospital stay”?

Yes, the sentence has been modified line 235.

  1. L161,2 and L181,2
    How were the results calculated?

L261,2: Clarified. (Line 269-70) It represents a reduction of 0.05 blood test per hospitalization day. In other terms, it represents a reduction of one blood test every two hospitalizations considering a length of stay of 10 days for each hospitalization (0.05 * 2 * 10 = 1 blood test)

  1. Table1: The P-value for the "≥65 years old" is not presented.

Added (Table 1).

  1. L268-270
    The present study considered the reduction of low-value care. The authors discussed site-specific high-value care, however, is this a discussion based on the present study?

High value care curriculum objectives from the Choosing wisely perspective are to decrease low value intervention such as those reported in our study.

Round 2

Reviewer 2 Report

COMMENTS TO AUTHORS

The authors made careful revisions to the points raised, and the manuscript is implemented. However, the following academic issues remains to be addressed.

  • The objective of the study was to limit the new prescription of low-value interventions and the authors excluded data with prior PPI or BZD prescription. Choosing Wisely campaigns proposed that “Don’t maintain long-term Proton Pump Inhibitor (PPI) therapy for gastrointestinal symptoms without an attempt to stop/reduce PPI at least once per year in most patients.” (Born, et al. Gen Fam Med 2019;20:9-12.). Inherently, it is appropriate for the evaluation to include not only new prescriptions of drugs, but also changes in the duration and dosage of those drugs. Authors extracted from EMR for convenience data collection. However, this is a single-center observational study, and detailed and appropriate data collection should be possible.
  • Impact of the Intervention: Seasonal variation in prescriptions for BZD and PPI should implied the possibility of seasonal variation in diagnosis on hospital admission. Therefore, diagnosis on hospital admission may be the potential confounder. However, The authors adapted age, sex, patient clinical complexity level and length of stay, rather than the diagnosis on hospital admission as confounders to be adjusted in this study. Confounding factors should be selected that are relevant to the outcome, and the rationale and justification for the selection should be clearly stated in the paper.
  • The calculation of blood tests and cost reduction rates may be presented as a reference to assist the reader's understanding, but the number of hospitalization days used in the both calculation is different, and the usefulness of the calculation results in the clinical practice and the basis for their presentation are not clear.

Author Response

  • The objective of the study was to limit the new prescription of low-value interventions and the authors excluded data with prior PPI or BZD prescription. Choosing Wisely campaigns proposed that “Don’t maintain long-term Proton Pump Inhibitor (PPI) therapy for gastrointestinal symptoms without an attempt to stop/reduce PPI at least once per year in most patients.” (Born, et al. Gen Fam Med 2019;20:9-12.). Inherently, it is appropriate for the evaluation to include not only new prescriptions of drugs, but also changes in the duration and dosage of those drugs. Authors extracted from EMR for convenience data collection. However, this is a single-center observational study, and detailed and appropriate data collection should be possible.

We thank the reviewer for this interesting remark. We totally agree the CW campaign suggests to deprescribe the PPI or to reduce the PPI dosage in most patients. In Switzerland, the CW campaign integrated this item in its top five list for ambulatory setting but not for hospital setting, as PPI cannot be interrupted abruptly without a risk of rebound effect. We recently published a study in ambulatory setting using the same kind of design. This study revealed that a quality circle intervention with individual provider feedback and peer comparison among primary care practices resulted in lower rates of PPI prescription (ref Kherad et al doi: 10.1007/s11606-021-06624-9)

For the present study taking place in hospital setting, we purposely limit the new prescription of low-value intervention as the objective was mainly to modify the behavior of our physicians by limiting the action bias which is a common cognitive bias among physicians.

Furthermore, we did not extract these data only for convenience. The dosage modification and the clinical circumstance of prescription were not available in the data extracted from our EMR. Even if we agree it is a limitation of our study that preclude conclusions on deprescription or dosage modification, this increases the feasibility to conduct this intervention in other hospitals that may have the same issues for the access of data.

From our perspective, the physicians working in hospital setting should play a role of sentinel and inform the GP working in ambulatory setting about inappropriate treatments that should be weaned. Again, deprescription is a very challenging process, but we plan to launch a new study using a software (Medsafer) to help physician to deprescribe progressively BZD or PPI in ambulatory setting.

  • Impact of the Intervention: Seasonal variation in prescriptions for BZD and PPI should implied the possibility of seasonal variation in diagnosis on hospital admission. Therefore, diagnosis on hospital admission may be the potential confounder. However, the authors adapted age, sex, patient clinical complexity level and length of stay, rather than the diagnosis on hospital admission as confounders to be adjusted in this study. Confounding factors should be selected that are relevant to the outcome, and the rationale and justification for the selection should be clearly stated in the paper.

We do agree that there is a possible seasonal variation in diagnosis on hospital admission. However, as mentioned in the manuscript, the pre- and post-intervention periods were defined so that the same calendar months were included to avoid any bias relative to seasonal impacts on patient diagnoses while ensuring a comparable experience of residents.  Furthermore, the detailed data on diagnosis on hospital admission were not accessible from data extracted from our EMR. Only data on age, sex, patient clinical complexity level and length of stay were accessible.

  • The calculation of blood tests and cost reduction rates may be presented as a reference to assist the reader's understanding, but the number of hospitalization days used in the both calculation is different, and the usefulness of the calculation results in the clinical practice and the basis for their presentation are not clear.

We corrected the manuscript to use the same length of stay reference than for the costs analyses on blood tests (9 days). We furthermore detailed the calculation to facilitate the reader understanding. (Lines 244-6).

Should the Editor agree the presentation is not clear enough, we are willing to remove this sentence: “This difference is almost equivalent to a reduction of one blood test every two hospitalizations considering our average length of stay of 9 days (-0.05*2*9 = -0.9)”.